# Association between *NLRP3 rs10754558* and *CARD8 rs2043211* Variants and Susceptibility to Chronic Kidney Disease

**DOI:** 10.3390/ijms24044184

**Published:** 2023-02-20

**Authors:** Antonella La Russa, Danilo Lofaro, Alberto Montesanto, Daniele La Russa, Gianluigi Zaza, Simona Granata, Michele Di Dio, Raffaele Serra, Michele Andreucci, Renzo Bonofiglio, Anna Perri

**Affiliations:** 1Department of Health Sciences, Magna Graecia University, 88100 Catanzaro, Italy; 2de-Health Lab, Department of Mechanical, Energy, Management Engineering, University of Calabria, 87036 Rende, Italy; 3Department of Biology, Ecology and Earth Sciences, University of Calabria, 87036 Rende, Italy; 4Section of Preclinical and Translational Pharmacology, Department of Pharmacy, Health and Nutritional Sciences, University of Calabria, 87036 Rende, Italy; 5Nephrology and Dialysis Unit, Department of Medical and Surgical Sciences, University of Foggia, 71121 Foggia, Italy; 6Department of Surgery, Division of Urology, SS Annunziata Hospital, 87100 Cosenza, Italy; 7Department of Medical and Surgical Sciences, Magna Graecia University, 88100 Catanzaro, Italy; 8Kidney and Transplantation Research Center, SS Annunziata Hospital, 87100 Cosenza, Italy; 9Department of Experimental and Clinical Medicine, Magna Graecia University, 88100 Catanzaro, Italy

**Keywords:** NLRP3 inflammasome, Chronic Kidney Disease (CKD), end-stage renal disease (ESRD), NLRP3, CARD8 polymorphisms, inflammation, fibrosis

## Abstract

Nod-like receptor protein 3 (NLRP3) is a multi-protein complex belonging to the innate immune system, whose activation by danger stimuli promotes inflammatory cell death. Evidence supports the crucial role of NLRP3 inflammasome activation in the transition of acute kidney injury to Chronic Kidney Disease (CKD), by promoting both inflammation and fibrotic processes. Variants of NLRP3 pathway-related genes, such as NLRP3 itself and CARD8, have been associated with susceptibility to different autoimmune and inflammatory diseases. In this study, we investigated for the first time the association of functional variants of NLRP3 pathway-related genes (*NLRP3-rs10754558*, *CARD8-rs2043211*), with a susceptibility to CKD. A cohort of kidney transplant recipients, dialysis and CKD stage 3–5 patients (303 cases) and a cohort of elderly controls (85 subjects) were genotyped for the variants of interest and compared by using logistic regression analyses. Our analysis showed a significantly higher G allele frequency of the NLRP3 variant (67.3%) and T allele of the CARD8 variant (70.8%) among cases, compared with the control sample (35.9 and 31.2%, respectively). Logistic regressions showed significant associations (*p* < 0.001) between NLRP3 and CARD8 variants and cases. Our results suggest that the *NLRP3 rs10754558* and *CARD8 rs2043211* variants could be associated with a susceptibility to CKD.

## 1. Introduction

Chronic inflammation is recognized as both a cause and a consequence of glomerular and tubulointerstitial damage occurring in Chronic Kidney Disease (CKD), leading to End-Stage Renal Disease (ESRD) [1,2,3]. The inflammation promoted by many types of kidney injury results from a complex network of interactions between renal parenchymal cells and resident/recruited immune cells, which triggers the secretion of inflammatory and profibrotic mediators, driving the fibrotic process [4]. Renal inflammation involves the innate immune system, which includes different receptors that activate pro-inflammatory signalling pathways. Among them, the NOD-like receptors form caspase-activating platforms, named inflammasomes, which have been recognized as critical mediators of inflammation. The best-understood inflammasome is the NOD-, LRR- and pyrin domain-containing protein 3 (NLRP3), which forms a complex with the adaptor protein ASC and the caspase-1 protease. Upon different stimuli, NLRP3 generates a multiprotein inflammasome complex whose activation leads to the conversion of pro-inflammatory interleukin-1β and −18 into their mature forms [5,6]. In turn, these cytokines provoke the expression and secretion of other proinflammatory cytokines, as well as an increase in the proliferation and cytotoxic activity of NK and CD8 + T cells, which are implicated in several autoimmune diseases [7,8,9]. In its resting state, NLRP3 activation by oligomerization is attenuated by the adaptor protein Caspase Recruitment Domain-containing family 8 (CARD8), which decreases IL-1β production by reducing the activation of nuclear factor-κB (NF-κB) and caspase-1, leading to the suppression of innate immune responses and inflammation [10,11]. Experimental models and clinical studies have demonstrated that the NLRP3 inflammasome mediates several mechanisms of CKD onset and progression, sustaining the sterile renal chronic inflammation observed in CKD through the regulation of proinflammatory cytokines, promoting tubule-interstitial injury, glomerular diseases, renal inflammation and fibrosis pathways [12,13,14,15,16]. Moreover, recent studies have described the inflammasome-independent functions of NLRP3 in non-immune cells of the kidney, which are strongly implicated in CKD pathogenesis [13,17]. The strength of these pieces of evidence has led to the development of several studies that focused on targeted therapeutics for up/downstream substances/effectors/factors of inflammasomes in kidney diseases, although their clinical efficacy still needs to be validated [18,19,20].

During recent decades, genetic studies highlighted the relationship between genetic variants of *NLRP3* and *CARD8* and the susceptibility to autoimmune and inflammatory diseases [21,22,23]. So far, about 60 Single Nucleotide Polymorphisms (SNPs) in the *NLRP3* gene have been identified and functional studies have revealed that some of them may lead to changes in its function, increasing (gain-of-function) or decreasing (loss-of-function) the activation of the inflammasome and the levels of IL-1β [21]. *NLRP3* is located in chromosome region 1q43–q44 and contains 3 kb upstream of the transcription start site (exons and introns) and 2 kb downstream of the stop codon (37,953 kb total). The *rs10754558* variant, located in the 3′-untranslated region (3′-UTR) of the *NLRP3* gene, has a certain impact on the stability of mRNA, as the risk allele contributes to mRNA stability more than the other allele. Functional analyses of *NLRP3-rs10754558*, performed on THP-1 cells, indicated that the higher expression of *NLRP3* mRNA was a component of the pathologic mechanisms leading to functional impairment [22]. *rs10754558* was predicted to localize in promoter histone markers, enhancer histone markers, DNase hypersensitivity and motif-changed (Foxo) and -affected bound proteins such as GA-binding protein (GABP) and CCCTC-binding factor (CTCF) [24]. Furthermore, *rs10754558* was associated with NLRP3 expression and identified as an expression quantitative trait locus (eQTL) in artery–breast–mammary tissue [25]. The *rs10754558* variant increases the expression of components of the NLRP3-IL-1β signalling pathway, as reported in the study of Zhang et al., showing an increased expression of IL-1β mRNA and serum IL-1β levels in patients carrying the GG genotype of *rs10754558* affected by primary gouty arthritis [26].

CARD8, a member of the Caspase recruitment domain (CARD) family, is encoded by the *CARD8* gene located on chromosome 19q13.33. The caspase recruitment domain contains a conserved homology domain that can mediate the protein–protein interactions among key apoptotic signalling molecules and participates in the nuclear factor kappa-B (NF-κB) signalling pathway via mediating the interactions of components in the upstream part of the pathway [27]. CARD8 is found to regulate the inflammatory reaction by inhibiting NF-κB signalling, which plays an important role in the immune and inflammatory responses and in apoptosis [28,29]. Regarding *CARD8* genetic variants, the polymorphism more extensively studied is *rs2043211*, which is associated with an increased risk of gout in Chinese and European populations [30,31] and with ischemic stroke [32]. It is a nonsense variant that comprises a T to A transition, resulting in a severely truncated protein that is unable to act as a negative regulator of the NLRP3 inflammasome [11,22,33]. *rs2043211* was predicted not to localize in enhancer histone markers and was associated with NLRP3 expression and identified as an eQTL in whole blood, thyroid and adipose visceral tissue [24,25].

Although the role of NLRP3 inflammasome activation in renal damage has been extensively studied, to the best of our knowledge only a few studies have investigated the association of *NLRP3* and *CARD8* genetic variants with a susceptibility to kidney disease. The clinical study of Tsetsos et al. showed that common variants in *CARD8* exerted a protective role against the development of diabetic nephropathy [34]. Moreover, Dessing et al. reported that variants in the *NLRP3* gene affect the risk of acute rejection following kidney transplantation [35]. Finally, Arbiol-Loca et al. investigated the relationship between the *ANRIL* SNP *rs10757278* and *CARD8* SNP *rs2043211* and the occurrence of cardiovascular events in a large cohort of renal transplant recipients, finding that patients carrying four risk alleles (*ANRIL* GG and *CARD8* AA), had 2.3-fold higher risk than carriers of any other genotype [36].

In this study, we investigated the potential association between the genetic variability of the functional variants *NLRP3-rs10754558* and *CARD8-rs2043211* and the susceptibility to CKD in a cohort of kidney transplant recipients (KTRs), dialysis and CKD stage 3–5 patients.

## 2. Results

### 2.1. Clinical Characteristics of Cases and Control Subjects

A total of 303 cases (143 KTRs, 126 dialysis and 34 CKD stage 3–5 patients) and 85 control subjects were recruited. Table 1 presents the clinical characteristics of the study subjects. Compared with the control group, cases were younger and, as expected, presented significant differences in the average serum levels of urea, uric acid, hemoglobin (Hgb), ferritin and C-Reactive Protein (CRP). Other significant differences were observed in serum levels of serum phosphate, cholesterol, triglycerides and albumin. No differences were found in the frequency of type 2 diabetes, whereas more cases were affected by hypertension. The primary cause of CKD was a Glomerular disease in KTRs (66.4%), Glomerular and Diabetic Nephropathy in the CKD stage 3–5 group (41.2%), while it was mostly unknown (64.3%) in dialysis patients (Appendix A).

### 2.2. Association of NLRP3 and CARD8 Gene Polymorphisms with CKD and ESRD

Both SNPs analysed were in the Hardy–Weinberg equilibrium in the control group (*p* = 0.67 for *rs10754558*, *p* = 0.21 for *rs2043211*). Significant differences in genotype (*p* < 0.001 for both variants) and allele (*p* < 0.001 for both variants) frequencies were found between control and cases. After adjustment for age, gender, diabetes and hypertension, logistic regression analysis showed significant associations between both *rs10754558* and *rs2043211* and CKD/ESRD (Table 2).

### 2.3. CRP Levels According to Genotype

To evaluate the possible association between the genetic variability of the analysed polymorphisms and the inflammation status of the KTR and dialysis groups of patients, we compared the CRP levels across the different genotypes of two groups. As shown in Figure 1, in KTRs, CRP levels were significantly higher in the rs10754558 GG genotype (median 6.81 mg/L, IQR 4.42–12.2) vs. CG (2.8, 0.9–5.2; *p* = 0.001) and CC (1.4, 0.8–2.9, *p* < 0.001), as well as in the rs2043211 TT genotype (5.86, 3.05–12.1) vs. AT (2.8, 0.65–8.42; *p* = 0.01) and AA (1.35, 0.39–4.5, *p* < 0.001). In the KTR group, no association was found between different genotypes and graft function in terms of eGFR (Appendix A) nor with the clinical diagnosis of Chronic Kidney Transplant Rejection (data not shown).

## 3. Discussion

The evidence emerging from our study suggests that the variants of *NLRP3*, *rs10754558* and *CARD8*, *rs2043211*, could be associated with CKD susceptibility.

Experimental data reported by studies performed using different models of kidney injury, such as chronic glomerulonephritis [37,38,39,40], diabetic nephropathy [41,42], Lupus Nephritis [43], crystalline nephropathy [44,45,46,47] and hypertensive nephropathy [48,49,50], have shown that NLRP3 inflammasome activation participates in the underlying pathogenetic events involved in the onset and progression of kidney damage, regardless of the aetiology of kidney disease. The contribution of NLRP3 machinery activation in the pathogenetic events affecting renal tissue, leading to CKD/ESRD, have been attributed to different molecular mechanisms, occurring through both canonical and non-canonical pathways [51]. In fact, the NLRP3 inflammasome not only mediates the inflammatory response but is also associated with myofibroblast differentiation during renal fibrosis, occurring through a crosstalk with the TGF-β/Smad signalling pathway [52], as also demonstrated by the significantly elevated expression levels of NLRP3 and caspase-1 in the kidneys of CKD patients [12]. The pathogenetic role of NLRP3 inflammasome activation in different kidney diseases is further supported by in vitro and in vivo evidence showing that the inhibition of NLRP3 signalling pathways often alleviates kidney injury, making the inflammasome a potential target in the treatment of renal disease [53]. In this *scenario*, in patients affected by kidney disease the presence of functional SNPs of NLRP3 pathway-related genes, alone or in combination, could influence the activation of the NLRP3 machinery, strongly contributing to increase/sustain renal inflammation and fibrosis, leading to CKD/ESRD. Interestingly, our genotyping results revealed a significantly higher frequency of the *rs10754558* G allele, as well as of the stop codon of *CARD8*, not only with respect to our control sample but also when compared to those reported in the European population (https://www.ncbi.nlm.nih.gov/snp/rs10754558; https://www.ncbi.nlm.nih.gov/snp/rs2043211, accessed on 15 September 2022) [54,55]. We are aware that the lack of measurement of IL-1β serum levels and the monocyte expression of NLRP3 components represents a limitation of the study. However, we observed higher CRP levels in patients carrying the GG genotype of *NLRP3rs10754558* and the stop allele of *rs2043211*, suggesting that the individuals carrying both SNPs have a higher proinflammatory profile.

Different studies have reported the association of *NLRP3* polymorphisms with autoimmune, inflammatory, cardiovascular and metabolic diseases [21,56,57,58], but few reports have focused on the association of NLRP3 pathway-related gene variants with kidney diseases. The clinical study of Cruz et al. reported that the *rs10754558* G allele was associated with lupus nephritis, but not with the risk of systemic lupus erythematosus development, suggesting that this variant could be directly involved in renal damage [59]. Conversely, Ehtesham et al. recently reported a significant association between the GG genotype and the G allele of *rs10754558* with increased risk of systemic lupus erythematosus as well as with the severity of disease activity, including renal involvement [60]. Furthermore, although it has been postulated that the *rs10754558* variant could be related to insulin resistance and increased risk of type 2 diabetes mellitus, no association with the risk of diabetic nephropathy was investigated [58,61]. Recently, the evidence of Tsetsos et al. suggested that three common *CARD8* variants, including *rs2043211*, confer decreased risk for diabetic nephropathy in patients affected by type 2 diabetes mellitus, although the authors declared that their preliminary observations would require validation in larger cohorts [34]. Regarding the role of NLRP3-related genetic variability, in kidney transplantation, to date, only one study reported that *rs35829419* in the donor and *rs6672995* in the recipient, both located in the *NLRP3* gene, contribute to the risk for biopsy-proven acute rejection, mainly within the first year after transplantation. No correlations were observed with delayed graft function, primary non-function, graft, or patient survival. Moreover, the authors reported that the genotypic distributions of *NLRP3* polymorphisms among graft donors and recipients were comparable with those observed for the general/Caucasian population [35]. In our study, we observed that neither of the two variants correlated with the occurrence of delayed graft function or primary non-function, although we could not investigate the correlation with the risk of acute rejection (data not shown).

Collectively, our results suggest that in patients affected by acquired kidney diseases, the genotyping for the above-reported SNPs at the time of diagnosis could be useful to better personalize the therapy. First, an anti-inflammatory and anti-oxidant nutritional approach should be strongly recommended for these patients, as it is likely that the patients carrying these SNPs could have a higher pro-inflammatory and pro-fibrotic profile. Certainly, the pharmacological inhibition/mitigation of NLRP3 inflammasome activation should represent the better approach mainly because, despite the fact that the treatment of CKD has made great progress, drugs to protect the kidney and delay the progress of CKD are still limited. However, although many kinds of biological inhibitors against the NLRP3 inflammasome have been developed, at present, their efficacy and safety for kidney diseases have not been defined [53].

## 4. Materials and Methods

### 4.1. Study Population

The patient sample consisted of KTRs followed between January and December 2017 at the Annunziata Hospital of Cosenza (Italy), and KTRs followed between January and March 2014 at the University Hospital of Verona (Italy), together with dialysis and CKD stage 3–5 (defined as eGFR < 60 mL/min/1.73 m^2^) patients followed in Cosenza during the first three months of 2018. The eGFR was calculated using the CKD-EPI formula [62]. Patients with a genetic–hereditary disease were excluded from the case sample.

The control sample was collected within the framework of several recruitment campaigns carried out for monitoring the quality of aging in the whole Calabria Region from 2010 onwards. Each subject was recruited after a complete multidimensional geriatric assessment with detailed clinical history, including anthropometric measures and common clinical haematological tests. All subjects had an eGFR > 60. White blood cells from blood buffy coats were used as a source of DNA. The socio-demographic characteristics of the control sample have been previously reported [63].

All procedures performed in this study were in accordance with the ethical standards of the institutional and national research committee and with the 1964 Helsinki Declaration and its later amendments or comparable ethical standards. The overall sample set enrolled provided written informed consent that, as guarantor, is retained by the corresponding author. No research ethics approval was obtained, since these data derive from routine clinical activity.

### 4.2. SNP Genotyping

Genomic DNA was extracted from peripheral blood leukocytes using the Wizard Genomic DNA Purification kit (Promega^®^, Madison, WI, USA) and quantified by spectrophotometry at 260 nm, using an Eppendorf BioSpectrometer. Genotyping was carried out by Polymerase Chain Reaction (PCR) and digestion reaction using previously reported restriction enzymes [22].

### 4.3. Statistical Analysis

Continuous variables were presented as mean ± SD if normally distributed or otherwise as a median (IQR). Categorical variables were presented as nr. (%). Two group differences were tested using Student’s *t*-tests, Mann–Whitney U test, or Pearson’s χ^2^ test as appropriate. Three group differences were tested with ANOVA test and Tukey’s HSD test for pairwise comparison after log transformation of non-normal variables. Hardy–Weinberg equilibrium was tested by χ^2^ test. Logistic regression analyses were performed to calculate genotype and allele Odds Ratio (OR) using age, sex, diabetes and hypertension as covariates. A *p*-value < 0.05 was considered statistically significant. All statistical analyses were performed using R 4.1.2 and relative packages [64,65,66,67].

## 5. Conclusions

The findings emerging from our study suggest that the genetic variants *rs10754558* and *rs2043211* could be associated with CKD, regardless of the aetiology of renal disease. Undoubtedly, the presumed association must be confirmed in a larger CKD/ESRD population, especially if genetic screening is to be used as a tool for predicting the risk of CKD progression and whether the pharmacological NLRP3 pathway inhibition will be included in the therapeutic armamentarium against renal diseases.

## Figures and Tables

**Figure 1 ijms-24-04184-f001:**
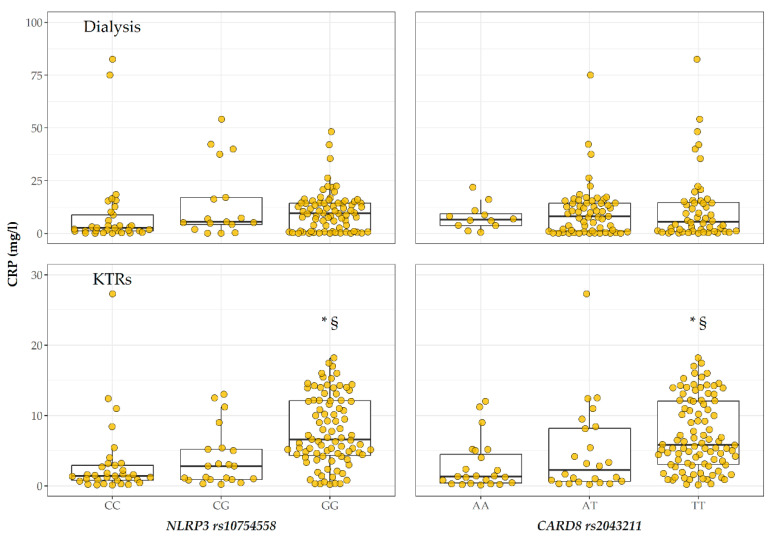
CRP levels by rs10754558 and rs2043211 genotype in Dialysis (upper panels) and KTRs (lower panels) patients. * *p* < 0.05 vs. CC/AA; § *p* < 0.05 vs. CG/AT.

**Table 1 ijms-24-04184-t001:** Clinical characteristics of cases and control subjects.

	Controls (n = 85)	Cases (n = 303)	*p*
Age (years)	72.06 ± 6.19	62.10 ± 14.88	<0.001
Sex (males)	51 (60.00)	199 (65.68)	0.402
Glucose (mg/dL)	109.06 ± 39.61	105.45 ± 38.44	0.455
Urea (mg/dL)	32.46 ± 9.43	54.84 ± 30.60	<0.001
Uric Acid (mg/dL)	4.63 ± 1.39	5.92 ± 1.50	<0.001
Ca (mg/dL)	9.45 ± 0.50	9.43 ± 0.83	0.845
Ph (mg/dL)	3.39 ± 0.58	4.01 ± 1.52	<0.001
Total Cholesterol (mg/dL)	206.38 ± 39.88	180.69 ± 48.39	<0.001
Triglycerides (mg/dL)	131.44 ± 65.74	176.66 ± 101.63	<0.001
HDL Cholesterol (mg/dL)	59.02 ± 16.37	49.51 ± 17.60	<0.001
LDL Cholesterol (mg/dL)	121.04 ± 34.21	95.22 ± 40.24	<0.001
Ferritinemy (ng/mL)	72.00 (42.00–161.00)	135.00 (45.50–391.50)	<0.001
CRP (mg/L)	2.46 (1.56–4.79)	5.18 (1.10–12.23)	0.01
Serum Albumin (gr/dL)	3.95 ± 0.32	3.72 ± 0.62	<0.001
Hgb (gr/dL)	14.21 ± 1.54	12.37 ± 1.90	<0.001
Type 2 Diabetes	13 (15.29)	63(20.79)	0.330
Hypertension	49 (57.65)	209 (68.98)	0.068

**Table 2 ijms-24-04184-t002:** Genotype and allele frequencies of NLRP3 rs10754558 and CARD8 rs2043211 in cases and control subjects.

	Controls (n = 85)	Cases (n = 303)	AOR (95% CI)	*p*
*NLRP3 rs10754558*				
CC	39 (45.88)	77 (25.41)	1	
CG	31 (36.47)	44 (14.52)	0.64 (0.32–1.24)	0.18
GG	15 (17.65)	182 (60.07)	6.04 (3.05–12.53)	<0.001
C allele	109 (64.12)	198 (32.67%)	1	
G allele	61 (35.88)	408 (67.33%)	3.56 (2.43–5.26)	<0.001
*CARD8 rs2043211*				
AA	44 (51.76)	39 (12.87)	1	
TA	29 (34.12)	99 (32.67)	4.41 (2.29–8.71)	<0.001
TT	12 (14.12)	165 (54.46)	14.34 (6.80–32.28)	<0.001
A allele	117 (68.82)	177 (29.21%)	1	
T allele	53 (31.18)	429 (70.79%)	5.06 (3.42–7.57)	<0.001

AOR: Adjusted Odds Ratio.

## Data Availability

The database is available on request.

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
