# Peer review of "Association between NLRP3 rs10754558 and CARD8 rs2043211 Variants and Susceptibility to Chronic Kidney Disease"

_ijms, 2023, doi:10.3390/ijms24044184_

Round 1
Reviewer 1 Report
SNPs are a rich field for inflammatory pathways understanding, especially among robustly described inflammasome genes as NLRP3. However, some major revisions must be taken to increase the richness of your findings.
1. TEXT FLOW AND STRUCTURE: Structure of the whole text should be revised. For example, introduction does not give enough context support to the reader, where only CARD8 SNP is satisfactory described at this session, other genes are cited with no explained connection to the main idea, e.g., ANRIL.I recommend that SNPs are fully described at the introduction, which could include its role as eQTL or sQTL, whether they form stop codons, the location of them in the protein and whether the site is susceptible to (de)methylation. Although the clinical outcomes of such SNPs are well cited during introduction, another paragraph of this session should consider the molecular activity of such proteins in this scenario, specially about fibrosis, which actively participate in kidney transplantation and CKD establishment. Here, a graphical abstract is welcome.
2. WRITING REVIEW: Some excerpts suggest lack of text revision, e.g., page 5 line 145 at “sustainedrenal” written together, or page 2 line 63-64 with a question mark in the middle of the sentence. 3. INFLAMMATORY PROFILE CONTEXT: Although you could not measure important cytokines for the inflammation, some of the analytes you have measured could help understand the difference found between CKD and KTR. I suggest that you explore analytes such as uric acid, urea and even cholesterol that could play important roles in those patients’ groups and show significant differences to your control group. 4. STUDY DESIGN: Study design is not clear. Please describe the reasons you consider to analyze both CKD and KTR patients, specially considering their inflammatory profiles since filtration status are totally different. 5. RESEARCH CONTRIBUTION: Suggestion of clinical and health approaches are not cited at the end of discussion. The research is not sufficiently conceptualized into its contribution to health care or industry fields.Author Response
SNPs are a rich field for inflammatory pathways understanding, especially among robustly described inflammasome genes as NLRP3. However, some major revisions must be taken to increase the richness of your findings.
- TEXT FLOW AND STRUCTURE: Structure of the whole text should be revised. For example, introduction does not give enough context support to the reader, where only CARD8 SNP is satisfactory described at this session, other genes are cited with no explained connection to the main idea, e.g., ANRIL.
We thank the reviewer for the accurate revision of our manuscript. As suggested, we better described NLRP3 SNP rs10754558 in the introduction. Regarding the ANRIL SNP rs10757278, in the revised manuscript we have better explained the role of its association with CARD8 SNP rs2043211 in renal transplant recipients.
- I recommend that SNPs are fully described at the introduction, which could include its role as eQTL or sQTL, whether they form stop codons, the location of them in the protein and whether the site is susceptible to (de)methylation. Although the clinical outcomes of such SNPs are well cited during introduction, another paragraph of this session should consider the molecular activity of such proteins in this scenario, specially about fibrosis, which actively participate in kidney transplantation and CKD establishment. Here, a graphical abstract is welcome.
We agree with the reviewer and accept the constructive suggestion made. Therefore, we have added to the introduction of the revised manuscript, by including informstion on the effect of both SNPs on transcription factor binding using HaploReg (http://pubs.broadinstitute.org/mammals/haploreg/haploreg.php), as well as their impact on gene expression in different tissues by the public GTEx (the Genotype-Tissue Expression) database (https://gtexportal.org/home/). There is no data in the literature showing that the sites are susceptible to (de)methylation. The possible involvement of these proteins in the fibrosis process has been included in the revised version of the Discussion section
- INFLAMMATORY PROFILE CONTEXT: Although you could not measure important cytokines for the inflammation, some of the analytes you have measured could help understand the difference found between CKD and KTR. I suggest that you explore analytes such as uric acid, urea and even cholesterol that could play important roles in those patients’ groups and show significant differences to your control group.
We have to apologize to the reviewers since during the submission the Table S1 uploaded file was incorrect. The new table has an additional column with the CKD stage 3-5 patients’ data. Evaluating other analytes as suggested, we observed that Urea and Uric Acid average values are high in all three patient subgroups, while only Dialysis and CKD patients showed significantly lower total cholesterol than controls. Moreover, we noted that CKD patients had median values of Ferritinemy and CRP similar to controls, and glucose values higher also compared to/with? KTRs and dialysis patients. We think that it is important to note that the differences observed between cases and controls were only partially influenced by the relatively small CKD patient group.
- WRITING REVIEW: Some excerpts suggest lack of text revision, e.g., page 5 line 145 at “sustainedrenal” written together, or page 2 line 63-64 with a question mark in the middle of the sentence.
We apologize for the typos present in the manuscript. We have revised the entire manuscript and removed them.
- STUDY DESIGN: Study design is not clear. Please describe the reasons you consider to analyze both CKD and KTR patients, specially considering their inflammatory profiles since filtration status are totally different.
To clarify the study design, we better explained the enrolled groups of patients in the Methods section. Since the aim of the study was to evaluate the possible association of the SNPs with chronic renal disease, we included KTRs (nr. 143) and dialysis patients (nr. 126), in other words patients who developed ESRD needing Renal Replacement Therapy during their life, as well as CKD stage 3-5 patients (nr. 34) with residual renal function
- RESEARCH CONTRIBUTION: Suggestion of clinical and health approaches are not cited at the end of discussion. The research is not sufficiently conceptualized into its contribution to health care or industry fields.
We thank the reviewer for this important suggestion. At the end of the discussion of the revised manuscript, we briefly reported the relevance of including genetic screening for both variants in the clinical management of patients affected by kidney diseases. In addition, as the pathogenetic role of NLRP3 activation in the onset and development of AKI and CKD is well known, we highlighted the potential contribution of NLRP3 pathway inhibition to delay CKD progression.
Reviewer 2 Report
Comments to manuscript ijms-2104541
This study analyzed the association of two variants in the NLRP3 and CARD8 genes with susceptibility to CKD. The role of the inflammasome in renal damage is an interesting topic that has recently attracted attention and requires further investigation. The manuscript is well written and substantiated. The authors report a strong association of genotypes rs10754558 G/G (OR=6), and rs2043211 T/T (OR=14). Considering that these are common variants, the OR values are outstanding for an association study. Although the authors note that these results should be confirmed in larger cohorts, caution with the results of association studies is necessary to decrease the risk of type I error.
Major comments:
1. The primary cause of CKD in the kidney transplant recipients (KTRs) and dialysis groups should be mentioned.
2. In the case of KTR group, what was the clinical status of the patients at the time of sample collection. It would be expected that patients with good allograft function would behave differently from those with events of rejection or allograft dysfunction... The authors could explain this patient group in greater detail since it is important for the comparison of genotypes with the state of inflammation (CRP values).
3. The greatest concern is in the large differences between the groups, considering that in both variants analyzed the minor alleles are not uncommon in the population. What was the quality control in genotyping? It is known that analysis of variants by RFLPs raises the risk of excess heterozygotes due to partial digestions.
4. Patients were recruited at different times and from different locations, particularly the transplant group. This can be source of population substructure by geographical distance. Have these population analyzed for other genetic markers? An analysis to rule out differences in gene frequencies due to population substructure would be useful.
Minor comments:
Check what is the official name of the NLRP3 gene?
Author Response
Reviewer 2:
This study analyzed the association of two variants in the NLRP3 and CARD8 genes with susceptibility to CKD. The role of the inflammasome in renal damage is an interesting topic that has recently attracted attention and requires further investigation. The manuscript is well written and substantiated. The authors report a strong association of genotypes rs10754558 G/G (OR=6), and rs2043211 T/T (OR=14). Considering that these are common variants, the OR values are outstanding for an association study. Although the authors note that these results should be confirmed in larger cohorts, caution with the results of association studies is necessary to decrease the risk of type I error.
Major comments:
- The primary cause of CKD in the kidney transplant recipients (KTRs) and dialysis groups should be mentioned.
The primary cause of CKD in KTRs and dialysis groups are reported in the supplementary Table 1. Furthermore, as suggested by the reviewer, we have mentioned them in the Results section.
- In the case of KTR group, what was the clinical status of the patients at the time of sample collection. It would be expected that patients with good allograft function would behave differently from those with events of rejection or allograft dysfunction. The authors could explain this patient group in greater detail since it is important for the comparison of genotypes with the state of inflammation (CRP values).
We evaluated the clinical status of KTR at sample collection. We found no association between eGFR and the analyzed SNPs (Figure S1) and also, as reported in the figure below, no association between eGFR, CRP and Chronic Rejection.
- The greatest concern is in the large differences between the groups, considering that in both variants analyzed the minor alleles are not uncommon in the population. What was the quality control in genotyping? It is known that analysis of variants by RFLPs raises the risk of excess heterozygotes due to partial digestions.
We thank the reviewer for this observation. The study was validated by DNA sequencing of heterozygotes to minimize the risk of genotyping errors due to partial digestions, and for this reason, no positive controls were used.
- Patients were recruited at different times and from different locations, particularly the transplant group. This can be source of population substructure by geographical distance. Have these population analyzed for other genetic markers? An analysis to rule out differences in gene frequencies due to population substructure would be useful.
The reviewer’s observations are very interesting. We did not analyze other genetic markers; however, we observed very similar genotypic frequency distribution of both SNPs among patients recruited at Verona Hospital (nr. 35) and Cosenza Hospital (nr. 108). This finding strongly suggests that the geographical distance is not relevant.
Minor comments:
Check what is the official name of the NLRP3 gene?
According to that reported by HGNC, the official name of NLRP3 gene is the following: NLR family pyrin domain containing 3. We have now reported it in the introduction of the revised manuscript.

Round 2
Reviewer 2 Report
I am still concerned about the genotyping. Although the authors mentioned in their response that DNA sequencing was performed to rule out partial digestion, it would be very useful for clarity reasons to show a gel photo after digestion and an electropherogram with the three genotypes for each polymorphism, at least as supplementary material.
Author Response
Response for reviewer 2_round_2
- I am still concerned about the genotyping. Although the authors mentioned in their response that DNA sequencing was performed to rule out partial digestion, it would be very useful for clarity reasons to show a gel photo after digestion and an electropherogram with the three genotypes for each polymorphism, at least as supplementary material.
We understand the correct reviewer's request. We confirm that at the time of experiments, the geneticist of our research group, performed the DNA sequencing of few patients, to confirm our findings. However, since it has been too long (2016-2017) unfortunately we did not find both the original electropherograms with the three genotypes and the gel images from computers.
We apologize for the lack.
Round 3
Reviewer 2 Report
The authors were not able to provide evidence to support their results.